# Novel Variants Linked to the Prodromal Stage of Parkinson’s Disease (PD) Patients

**DOI:** 10.3390/diagnostics14090929

**Published:** 2024-04-29

**Authors:** Marwa T. Badawy, Aya A. Salama, Mohamed Salama

**Affiliations:** 1Biology Department, School of Sciences and Engineering, The American University in Cairo, New Cairo 11835, Egypt; marwa-tawfik@aucegypt.edu; 2Applied Science, Windows and Web Experience, Microsoft, Cairo 11561, Egypt; aya_salama@aucegypt.edu; 3Institute of Global Health and Human Ecology (IGHHE), The American University in Cairo, New Cairo 11835, Egypt; 4Toxicology Department, Faculty of Medicine, Mansoura University, Mansoura 35516, Egypt; 5Global Brain Health Institute, Trinity College Dublin, D02 X9W9 Dublin, Ireland

**Keywords:** Parkinson’s disease (PD), the Parkinson’s Progression Markers Initiative (PPMI), VCF, neurodegenerative diseases

## Abstract

Background and objective: The symptoms of most neurodegenerative diseases, including Parkinson’s disease (PD), usually do not occur until substantial neuronal loss occurs. This makes the process of early diagnosis very challenging. Hence, this research used variant call format (VCF) analysis to detect variants and novel genes that could be used as prognostic indicators in the early diagnosis of prodromal PD. Materials and Methods: Data were obtained from the Parkinson’s Progression Markers Initiative (PPMI), and we analyzed prodromal patients with gVCF data collected in the 2021 cohort. A total of 304 participants were included, including 100 healthy controls, 146 prodromal genetic individuals, 21 prodromal hyposmia individuals, and 37 prodromal individuals with RBD. A pipeline was developed to process the samples from gVCF to reach variant annotation and pathway and disease association analysis. Results: Novel variant percentages were detected in the analyzed prodromal subgroups. The prodromal subgroup analysis revealed novel variations of 1.0%, 1.2%, 0.6%, 0.3%, 0.5%, and 0.4% for the genetic male, genetic female, hyposmia male, hyposmia female, RBD male, and RBD female groups, respectively. Interestingly, 12 potentially novel loci (MTF2, PIK3CA, ADD1, SYBU, IRS2, USP8, PIGL, FASN, MYLK2, USP25, EP300, and PPP6R2) that were recently detected in PD patients were detected in the prodromal stage of PD. Conclusions: Genetic biomarkers are crucial for the early detection of Parkinson’s disease and its prodromal stage. The novel PD genes detected in prodromal patients could aid in the use of gene biomarkers for early diagnosis of the prodromal stage without relying only on phenotypic traits.

## 1. Introduction

The early detection of Parkinson’s Disease is an important factor in successful intervention. One of the areas worth studying is the prodromal stage of PD. In this paper, we studied the Parkinson’s Progression Markers Initiative (PPMI) genetic data, trying to identify the novel variants that are more prevalent in this stage of the disease. Several new genes were identified that may be related to the prodromal stage of PD and can help as early markers of the disease.

Parkinson’s disease (PD) is a neurodegenerative disease that is pathologically defined as the death of dopaminergic neurons in the midbrain and the inclusion of Lewy bodies in the brain [1]. It has become obvious that PD has a prodromal stage, which is the period before the beginning of neurodegeneration without detection of motor signs by classical diagnosis. The basis of the nonmotor prodromal stage is that the pathological process cannot yet occur in the substantia nigra pars compacta (SNpc) [2]. The classical diagnosis of PD relies on the loss of mesodiencephalic dopaminergic (mdDA) neurons in the substantia nigra pars compacta (SNpc) and the development of Lewy bodies in some surviving neurons [3].

Early investigations focused on the role of genetic factors in Parkinson’s disease to identify rare mutations linked to familial disease [4]. Moreover, the past decade has shown the great role of genetics in sporadic disease [5]. The identification of novel variants and genes for the early diagnosis of the prodromal and/or PD stages is receiving increased amounts of attention [6]. Variant call format (VCF) is becoming a community standard for reporting variations in genetic data acquired from medical genetic diagnostics and research [7].

In this study, we analyzed gVCF data acquired from the Parkinson’s Progression Markers Initiative (PPMI) for healthy participants and prodromal PD patients. The PPMI is a comprehensive, international, and observational study designed to identify PD progression biomarkers, initiated by the Michael J. Fox foundation in 2010. All data integrated in this study can be found at the PPMI official website at www.ppmi-info.org, accessed on 1 April 2024 [8].

The prodromal PD subgroups involved in this study were genetic, RBD, and hyposmia. Sleep behavior disorder (RBD), a rapid eye movement (REM) disorder, is a parasomnia condition characterized by complex abnormal motor movements in the REM state during sleep [9]. RBD is mainly associated with abnormal movement behaviors, nightmares, and the loss of normal skeletal atonia in the REM state [10]. Moreover, hyposmia is an olfactory dysfunction that leads to a loss of smell ability and is the most common nonmotor symptom of PD [11].

This study was based on annotation, biological pathway, and disease association analyses of prodromal patients with PD. We detected the percentages and types of variation in each population with their percentile on each chromosome alongside the novel/existing variants percentages, as shown in Table 1. The gene lists were generated and are presented as gene-annotation network clusters and bar charts, which enabled us to detect novel genes with counts in prodromal stages identified recently in PD patients. Furthermore, the summary of the detected pathways and association disease analysis from DisGeNET and HPO datasets is illustrated in Table 2.

## 2. Materials and Methods

### 2.1. PPMI—Data Collection

The data were obtained from the Parkinson Progression Markers Initiative (PPMI). The PPMI is an international, multisite, prospective, and observational study investigating biomarkers for Parkinson’s disease (PD). The specific study methodology can be found at www.ppmi-info.org (accessed on 1 April 2024).

### 2.2. PPMI—Study Participants

The participants in the PPMI study met these criteria: were at least 30 years of age, presented two of three cardinal symptoms (bradykinesia, rigidity, or resting tremor), were diagnosed within 2 years before entering that study, were untreated for PD when entering that study, and had a deficit in dopamine transporters [12]. On the other hand, the healthy control participant criteria were the following: having no neurological disorder, no first-degree relative with PD, and a normal dopamine transporter (DAT) identified through single-photon emission computed tomography (SPECT) imaging by visual inspection [8]. Written informed consent was obtained from all the participants in the PPMI study, including for the genetic research part. The PPMI study was conducted under the ethical standards of the Helsinki Declaration of 1975.

### 2.3. Study Design

In this study, we considered the very recent cohort from the PPMI, the April 2021 cohort, which generally represents the prodromal PD population and healthy participants. A pipeline was developed for this work; gVCF files were used as input, and analyses were performed to convert gVCF files to VCF files, apply variant quality control, extract summaries of all variants in each chromosome, merge filtered VCF files and annotating variants, and finally carry out pathway analysis and disease association, as illustrated in Figure 1.

The following bioinformatics tools were installed: Genome Analysis Toolkit (GATK) [13], Variant Call Format tools (VCFtools) [7], Binary Calling Format tools (BCFtools), Sequence Alignment/Map tools (SAMtools) [14], Burrows–Wheeler Aligner (BWA) [15], Picard tools, and Variant Effect Predictor (VEP) [16]. These tools were used to perform the specified analyses within the pipeline. The BCFtools thresholds for filtering the variants were a quality score (QUAL) ≥30 and a read depth (DP) >20. 

These gVCF data consisted of 100 healthy controls (HCs) and 165 prodromal individuals, comprising 146 genetic, 21 with hyposmia, and 37 with RBD. The HCs were divided into 50 males and 50 females. The prodromal genetic group included 61 males and 85 females, the prodromal RBD group included 31 males and 6 females, and the prodromal hyposmia group included 14 males and 7 females. The participants were classified based on their age, and each age group consisted of 5 years, as shown in Figure 2.

## 3. High Percentage of Intronic and Intergenic Variants

The Ensembl Variant Effect Predictor (VEP) is a powerful tool for analyzing genomic variation in coding and noncoding regions. It also provides access to a very extensive collection of genomic annotations [16].

In this study, the VEP was used for annotation analyses and the detection of genomic variations and their associations. The highest two percentages of variation were detected for intronic and intergenic variants, as illustrated in Figure 3A. Similarly, 51.5% and 35.3% of the healthy control male and female populations, respectively, exhibited the same percentages of intron variants and intergenic variants. Among those in the prodromal genetic male population, 51.3% had intron variants and 35.6% had intergenic variants, while among those in the prodromal genetic female population, 51.2% had intron variants and 35.7% had intergenic variants. The prodromal RBD males and females had the same percentages of intron variants and intergenic variants (52% and 34.8%, respectively). Among the males with prodromal hyposmia, 51.8% had intron variants and 35% had intergenic variants; moreover, among the females with prodromal hyposmia, 51.9% had intron variants and 34.9% had intergenic variants.

The detection of variants in the intronic and intergenic regions is common across the entire human genome, as these noncoding regions make up half of the human noncoding genome and can play important regulatory roles [17]. The presence of intronic and intergenic variants in the studied healthy population and prodromal populations suggests that these variants are not specifically associated with prodromal PD. However, these findings likely represent background genetic variation.

A single-nucleotide variant (SNV), also called a single-nucleotide polymorphism (SNP), is a variant of a specific single nucleotide and occurs at a specific position in the genome. Moreover, SNVs are the most common type of genetic variation [18]. This clearly explains the high percentages of SNVs detected in all the healthy and prodromal populations presented in Figure 3B. All of the detected SNVs were ≥82% in all populations.

Deletion mutations were detected as the second highest number of elements after SNVs. The two highest percentages of deletions were detected in the RBD females and hyposmia females (7.9% and 7.8%, respectively). These deletion mutations may be associated with genetic factors involved in the development of hyposmia and RBD in individuals, specifically females. Moreover, these deletions could contain genes or regulatory regions relevant to olfactory function and sleep regulation. A sex difference could be related to the association between high deletion percentages and prodromal symptoms in females [19]. This could suggest potential sex-specific genetic risk factors for PD. Consequently, the two most common insertion mutations were detected in 7.9% of the RBD females and 7.8% of the hyposmia females, as shown in Figure 2B. The detection of insertions in the RBD and hyposmia populations suggested genetic variability within these groups. These high percentages of insertions could be associated with disease susceptibility or progression. Additionally, depending on their location within the genome, these insertions can have various functional consequences.

Chromosomes 1 and 2 are among the largest chromosomes in the human genome and contain a large number of genes and regulatory elements [20]. Therefore, these genes may represent a greater number of variants simply because of their size and gene density. High percentages of variants were detected in all the analyzed samples, including those of the healthy controls. The healthy control males had variant counts of 1,500,086 on chr1 and 1,423,662 on chr2, while the healthy control females had variant counts of 1,509,369 on chr1 and 1,441,362 on chr2 (Figure 3C). Consequently, it could be normal to find these variants in prodromal populations. However, chromosomes 1 and 2 may influence the biological processes relevant to PD, such as mitochondrial function, protein aggregation, the oxidative stress response, and neuroinflammation. Understanding how variants in these genomic regions affect the molecular pathways associated with PD is crucial for providing insights into disease mechanisms. Gene-annotation network cluster and pathway analyses are shown in Figure 4, Figure 5 and Figure 6.

All the input lines of the variants were processed, and the number of filtered variants was zero in all the populations. We used the filtered VCF file that we produced during the pipeline after the QC step. That table showed that novel variation percentages were detected in all the tested populations. The detection of novel genetic variations in healthy populations is a natural consequence of genetic diversity and the complexity of the human genome. As a result, the novel variation detected in the tested population may be natural compared to the percentage of healthy controls. However, the chromosomal location and type of variant, whether Indels or SNVs, could provide a deeper justification of whether these variants could be potentially related to prodromal PD or whether they are just natural variants.

## 4. Disease–Gene Network (DisGeNET) Detection in Prodromal PD Populations

The disease–gene network, known as DisGeNET, is a comprehensive knowledge base that integrates information on human disease-associated genes and variants from multiple sources [21]. This database was accessed through the GeneCodis website, and annotation was carried out through this tool [22]. Acquired hypogammaglobulinemia was detected in the heathy male population, in which 11 genes were associated with this disease. Acquired hypogammaglobulinemia is also known as secondary hypogammaglobulinemia and is a condition characterized by low levels of immunoglobulins (antibodies) in the blood. This condition can increase the risk of infections and can occur due to various factors, such as certain medications, underlying medical conditions, or environmental exposures [23]. Therefore, the detection of this disease in the healthy male population could be due to the age of the participants, as 49 of the participants were aged ≥45 years, as shown in Figure 4. On the other hand, 75 genes were detected to be related to dermatological disorders in the healthy female population. Many dermatological disorders, including eczema, psoriasis, acne, and others, involve multiple genes and have complex genetic architectures. Variations in these genes can influence susceptibility to these conditions, and the involvement of 75 genes may indicate a polygenic inheritance pattern. Each gene may have a small effect on the overall risk of developing dermatological disorders. Environmental factors also contribute to disease susceptibility, and exposures to allergens, irritants, pollutants, UV radiation, and microbial agents can interact with genetic predispositions to trigger or exacerbate skin conditions.

Pheochromocytoma and hypertriglyceridemia were detected in the prodromal genetic male population, with 19 and 15 genes, respectively. Pheochromocytoma (PCC) is a rare neuroendocrine tumor that arises from the adrenal glands and can also occur elsewhere in the sympathetic nervous system. It is characterized by the excessive production of catecholamines, such as noradrenaline and adrenaline, which can cause a variety of symptoms, including palpitations, hypertension, headache, anxiety, and sweating [24]. The detection of pheochromocytoma in the prodromal genetic PD population is rare and unusual but possible. While there is no known direct genetic link between PCC and PD, it is important to note that both conditions can be influenced by genetic predispositions, environmental factors, and complex interactions between various biological pathways. Additionally, several genes associated with PD may have other roles in different cellular processes beyond the central nervous system.

Hypertriglyceridemia is an elevated level of triglycerides in the blood and is a lipid abnormality associated with an increased risk of cardiovascular disease [25]. Hypertriglyceridemia is primarily influenced by lifestyle factors such as physical activity, diet, and obesity. Genetic factors can also play a role in lipid metabolism and contribute to elevated triglyceride levels. Investigating potential shared genetic predispositions between HTG and PD may provide insights into overlapping biological pathways or susceptibility genes. Furthermore, the dysregulation of lipid metabolism and glucose homeostasis has been implicated in the pathogenesis of PD. Emerging evidence suggests potential links between metabolic dysfunction, insulin resistance, and neurodegeneration in PD patients. Detecting hypertriglyceridemia in individuals with prodromal genetic PD in the male population may raise questions about underlying metabolic disturbances and their implications about disease progression.

Among the prodromal genetic female population, single seizures were detected, with 101 genes. These seizures can be triggered by fever (febrile seizures), head injury, metabolic disturbances, sleep deprivation, stress, alcohol, or drug withdrawal [26]. While PD primarily affects dopaminergic neurons in the brain, there is evidence to suggest that individuals with PD or in the prodromal stage may have an increased susceptibility to seizures compared to the general population. Genetic factors, including mutations in genes associated with both PD and epilepsy, could contribute to this increased risk. The involvement of 101 genes may indicate a polygenic or multifactorial basis for the seizure phenotype, with variations in multiple genes contributing to the risk of seizures.

Cone–rod synaptic disorder (CRSD) was detected in 13 genes of the prodromal RBD male population. CRSD is a rare genetic disorder characterized by dysfunction of the synaptic connections between the cone and rod photoreceptor cells in the retina. This leads to visual impairment, particularly affecting color vision, central vision, and visual acuity [27]. Additionally, RBD is a rapid eye movement behavior disorder characterized by the loss of muscle atonia during REM sleep, leading to the enactment of dreams through vocalizations and movements. While CRSD primarily affects the retina, several genes associated with retinal function may have broader roles in neuronal health and function. Variants in these genes could contribute to neurodegenerative processes in conditions such as PD.

Autosomal recessive primary microcephaly (MCPH) was detected in the prodromal RBD female population, with 22 genes. MCPH is a rare neurodevelopmental disorder characterized by significantly reduced head size (microcephaly) and intellectual disability. It is inherited in an autosomal recessive manner, meaning that both copies of the affected gene (one from each parent) must be mutated for the condition to manifest [28]. While MCPH primarily affects brain development, several genes associated with neurodevelopmental disorders may have broader roles in neuronal health and function. Genetic variations in these genes may contribute to neurodegenerative processes in conditions such as prodromal RBD.

Adjuvant arthritis was detected in the prodromal hyposmia male population, with 40 genes. In humans, a type of reactive arthritis occurs when the immune system reacts to a triggering event, such as an infection or exposure to certain substances. It typically presents with joint pain, swelling, and stiffness, similar to other forms of inflammatory arthritis. PD and autoimmune disorders such as rheumatoid arthritis (RA) have distinct etiologies, and there is a growing recognition of shared genetic susceptibility and environmental factors that may contribute to both conditions. However, when arthritis was detected, the adjuvant-induced arthritis in the male population with prodromal hyposmia, involving 40 genes, suggested a complex interplay of genetic and environmental factors. More importantly, this arthritis occurs at older ages, and all the hyposmic male patients were aged ≥60 years. On the other hand, familial Alzheimer’s disease (FAD) was detected in 99 prodromal hyposmia females. FAD is associated with mutations in specific genes, including the amyloid precursor protein (APP), presenilin 1 (PSEN1), and presenilin 2 (PSEN2) [29]. These mutations are typically inherited in an autosomal dominant manner, meaning that a single copy of the mutated gene is sufficient to cause the disease. While some genetic mutations may be associated with both Alzheimer’s disease and Parkinson’s disease, detecting FAD in a prodromal hyposmia female population involving 99 genes would need further in-depth investigations.

## 5. Detection of Human Phenotype Ontology (HPO) Data in Prodromal PD Populations

The HPO dataset is the Human Phenotype Ontology, which consists of phenotypic abnormalities encountered in human disease [30].

Hyperinsulinemia was detected in the healthy male population, with 120 genes. Hyperinsulinemia is a condition characterized by higher-than-normal levels of insulin in the blood. Insulin is a hormone, produced by pancreatic bet cells, that helps regulate glucose levels by facilitating the uptake of glucose into cells for energy or storage [31]. The detection of hyperinsulinemia in the male population of HCs may suggest underlying metabolic abnormalities or insulin resistance. Moreover, hyperinsulinemia can occur as a compensatory response to insulin resistance and can be influenced by various factors, such as diet, physical activity, genetics, and medications. This could also be justified by the older ages of the healthy male population, as they were ≥45 years old.

The HPO results revealed that all the female populations, including the healthy population, had the same gene network of autosomal dominant inheritance, with 1828 genes being involved. Autosomal dominant inheritance is a pattern of inheritance for a trait or disorder determined by genes located on autosomal chromosomes (non-sex chromosomes). In other words, a single copy of the mutated gene, inherited from one parent, is sufficient to express the trait or disorder. This means that individuals who inherit the mutated gene from either parent will exhibit the trait or disorder. Examples of disorders with autosomal dominant inheritance include Parkinson’s disease, Huntington’s disease, familial hypercholesterolemia, Marfan syndrome, neurofibromatosis type 1, and some other forms of familial Alzheimer’s disease [32].

Cerebral hemorrhage was detected in the prodromal genetic male population, with 62 genes. Cerebral hemorrhage is a medical condition characterized by bleeding within brain tissues. It occurs when a blood vessel within the brain ruptures, leading to the leakage of blood into the surrounding brain tissue. This bleeding can cause damage to brain cells and disrupt normal brain function [33]. In general, cerebral hemorrhage is not a common feature of prodromal PD. However, its detection in the prodromal genetic PD male population suggests a potential overlap or interaction between the genetic factors predisposing patients to PD and cerebrovascular events.

Respiratory insufficiency due to muscle weakness was detected in 79 genes in the prodromal RBD male population. In this condition, the muscles involved in breathing are unable to adequately perform their function, leading to impaired respiratory function. This can occur because of various underlying causes, including neurological conditions, neuromuscular disorders, or muscular dystrophies [34]. Detecting respiratory insufficiency due to muscle weakness in the prodromal RBD PD male population, with 79 genes, may suggest genetic predispositions or variants associated with neuromuscular or respiratory function. Moreover, respiratory insufficiency in PD patients is more commonly associated with factors such as upper airway obstruction, aspiration pneumonia, or respiratory muscle rigidity.

Prenatal maternal abnormalities were detected in the prodromal hyposmia male population, with 23 genes. Prenatal maternal abnormalities are not linked to maternal health, but they may also occur because of genetic conditions or mutations carried by the father, which can be transmitted to the fetus and influence prenatal development and health outcomes. Epigenetic modifications, such as DNA methylation patterns or histone modifications, can reflect prenatal environmental exposures or maternal health conditions [35]. Epigenetic changes could influence gene expression and neurodevelopmental processes relevant to the PD risk.

## 6. Online Mendelian Inheritance in Man (OMIM) Detection in Prodromal PD Populations

The OMIM database contains Mendelian Inheritance in Man. It is a comprehensive and authoritative compendium of human genes, genetic disorders, syndromes, and traits [36].

The Online Mendelian Inheritance in Man was the third phenotypic dataset to be used in this study to obtain a wider view of the whole of the three available phenotypic databases. Notably, none of the populations exhibited significant pathway or significant gene-network cluster annotations. However, one to four genes were detected from each detected pathway. In the HC male population, idiopathic pulmonary fibrosis (IPF) was detected, with four genes. IPF is a chronic and progressive lung disease characterized by scarring (fibrosis) of the lungs, leading to impaired lung function and difficulty breathing. The exact cause of IPF is unknown; however, IPF is believed to result from a combination of environmental factors, genetic predispositions, and abnormal wound healing processes in the lungs [37]. This could also be explained by the older age of the HC male population, as mentioned previously.

The reason why there was no significant pathway or gene network detected could be justified due to the rare Mendelian forms of PD. The known Mendelian forms of PD include certain monogenic forms caused by mutations in genes such as PARK2, SNCA, and LRRK2; these forms represent a small proportion of all PD cases, particularly those in the prodromal stage [38].

## 7. Novel Gene Detection in Prodromal PD Populations

Interestingly, 12 potentially novel PD loci, recently detected by Kim [1], were found to be present in prodromal populations. The 12 potentially novel loci were MTF2, PIK3CA, ADD1, SYBU, IRS2, USP8, PIGL, FASN, MYLK2, USP25, EP300, and PPP6R2. Table 3 shows the 12 novel genes and their counts in each prodromal population. This detection could lead to the use of these genes as potential genetic biomarkers for the early detection of prodromal PD patients.

## 8. Conclusions

Genetic composition plays a crucial role in the development of Parkinson’s disease and its prodromal stage subgroups. The novel PD genes detected in prodromal patients could aid in the use of gene biomarkers for early diagnosis of the prodromal stage without relying only on phenotypic traits. The network clusters of the prodromal populations showed how prodromal PD subgroups may be linked to a wide range of diseases and complications. This is mainly because PD-related genetic factors may have other functions beyond the nervous system that can result in other complications and illnesses throughout the human body. However, further clinical research is needed to provide in-depth information about the representative genetic results.

## Figures and Tables

**Figure 1 diagnostics-14-00929-f001:**
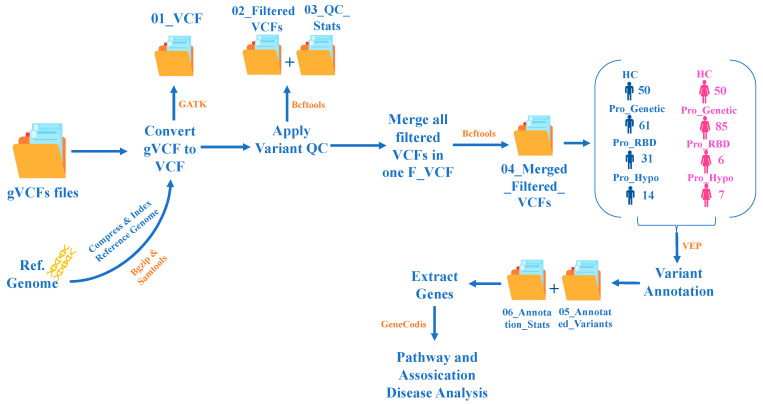
Pipeline workflow diagram for the genomic variation analysis. This pipeline takes gVCF files as input and performs the following analyses: convert gVCF to VCF, variant quality control, extract summary of all variants in each chromosome, merge VCF filtered files, variant annotation, pathway analysis, and disease association.

**Figure 2 diagnostics-14-00929-f002:**
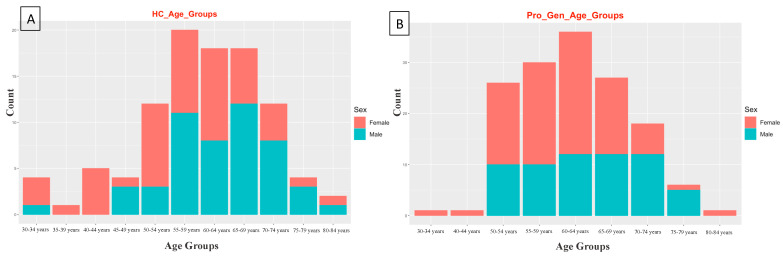
Bar plots for the age groups of the study participants. (**A**) Healthy control (HC) age groups ranging from 30 to 84 years. (**B**) Prodromal genetic (Pro_Gen) age groups ranging from 30 to 84 years. (**C**) Prodromal RBD (Pro_RBD) age groups ranging from 55 to 84 years. (**D**) Prodromal hyposmia (Pro_Hypo) age groups ranging from 60 to 84 years. Figure generated with R software, version 4.3.2.

**Figure 3 diagnostics-14-00929-f003:**
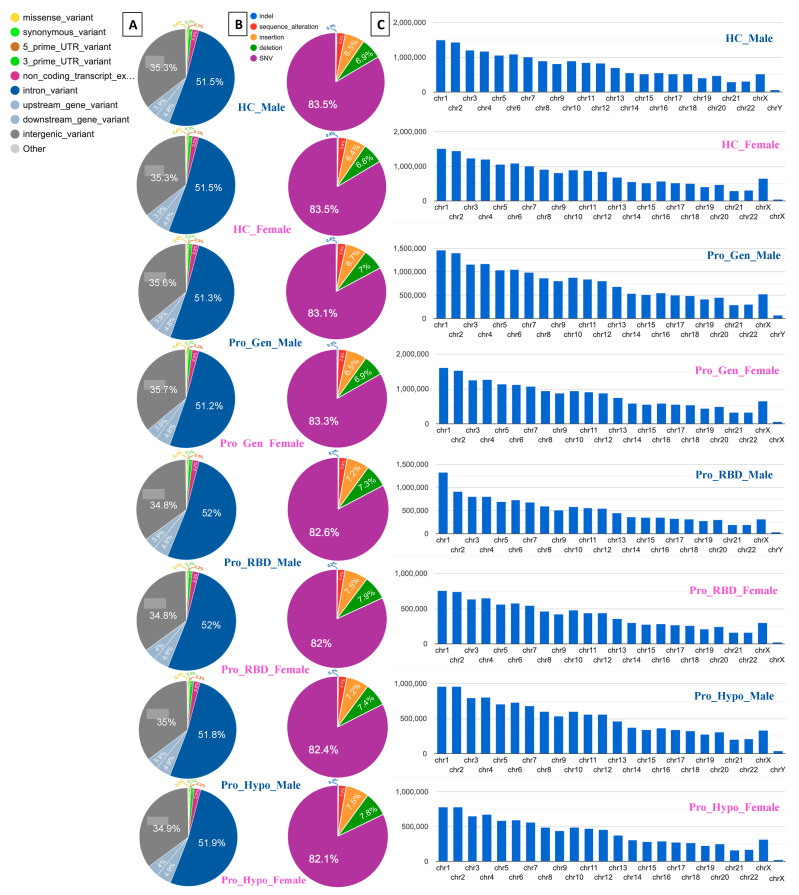
(**A**) Pie chart visualization of the most severe variant consequences, (**B**) pie chart visualization of the variant classes, and (**C**) bar plot visualization of variants by chromosome for each population. Pie charts and bar plots were generated using the ensemble VEP tool.

**Figure 4 diagnostics-14-00929-f004:**
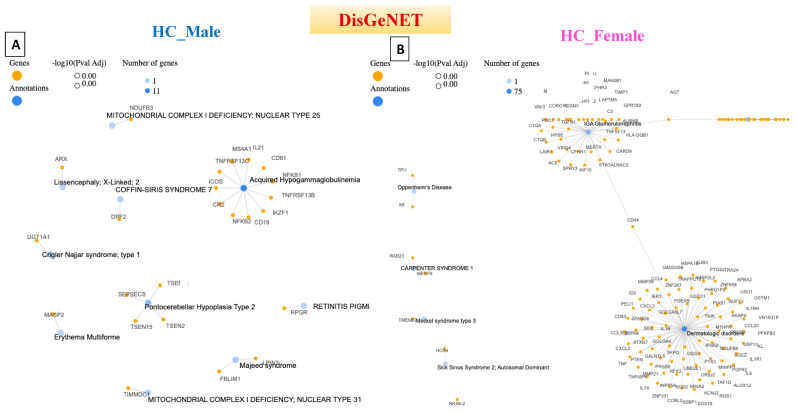
Network cluster connections based on gene pathway for the disease-gene network (DisGeNET). (**A**) Healthy Control male (HC_Male), (**B**) Healthy Control female (HC_Female), (**C**) Prodromal Genetic Male (Pro_Gen_Male), (**D**) Prodromal Genetic Female (Pro_Gen_Female), (**E**) Prodromal RBD Male (Pro_RBD_Male), (**F**) Prodromal RBD Female (Pro_RBD_Female), (**G**) Prodromal Hyposmia Male (Pro_Hypo_Male), and (**H**) Prodromal Hyposmia Female (Pro_Hypo_Female). Networks were generated with GeneCodis.4.

**Figure 5 diagnostics-14-00929-f005:**
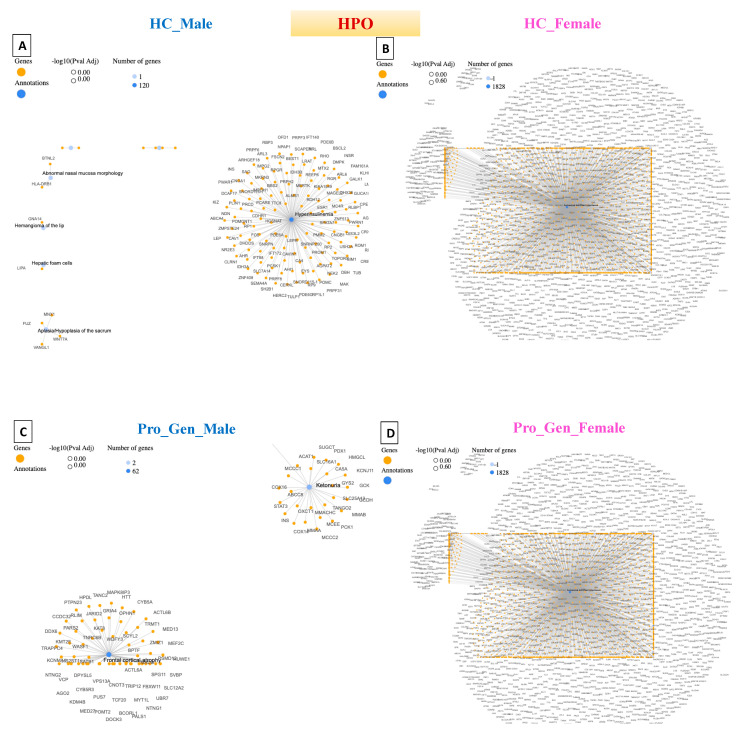
Network cluster connections based on gene pathways for the Human Phenotype Ontology (HPO). (**A**) Healthy Control male (HC_Male), (**B**) Healthy Control female (HC_Female), (**C**) Prodromal Genetic Male (Pro_Gen_Male), (**D**) Prodromal Genetic Female (Pro_Gen_Female), (**E**) Prodromal RBD Male (Pro_RBD_Male), (**F**) Prodromal RBD Female (Pro_RBD_Female), (**G**) Prodromal Hyposmia Male (Pro_Hypo_Male), and (**H**) Prodromal Hyposmia Female (Pro_Hypo_Female). Networks were generated with GeneCodis.4.

**Figure 6 diagnostics-14-00929-f006:**
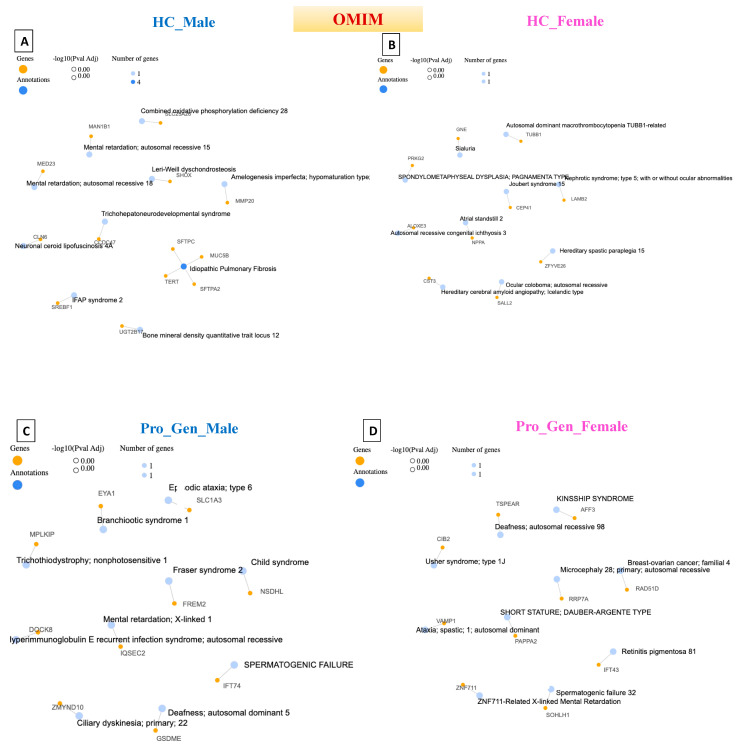
Network cluster connections based on gene pathways for Online Mendelian Inheritance in Man (OMIM). (**A**) Healthy Control male (HC_Male), (**B**) Healthy Control female (HC_Female), (**C**) Prodromal Genetic Male (Pro_Gen_Male), (**D**) Prodromal Genetic Female (Pro_Gen_Female), (**E**) Prodromal RBD Male (Pro_RBD_Male), (**F**) Prodromal RBD Female (Pro_RBD_Female), (**G**) Prodromal Hyposmia Male (Pro_Hypo_Male), and (**H**) Prodromal Hyposmia Female (Pro_Hypo_Female). Networks were generated with GeneCodis.4.

**Table 1 diagnostics-14-00929-t001:** General statistics of the variation analysis results for the healthy control (HC) (male and female), prodromal PD (genetic male and female), prodromal PD (RBD male and female), and prodromal PD (hyposmia male and female) populations.

Population	No. of Samples in Population	Lines of Input	Processed Variants	Novel/Existing Variants	Overlapped Genes	Overlapped Transcripts
HC_Male	50	18,191,327	18,191,327	168,623 (0.9%)/18,022,704 (99.1%)	61,987	250,277
HC_Female	50	18,387,115	18,387,115	182,458 (1.0%)/18,204,657 (99.0%)	61,663	249,831
Pro_Gen_Male	61	17,763,968	17,763,968	179,814 (1.0%)/17,584,154 (99.0%)	62,009	250,316
Pro_Gen_Female	85	19,371,326	19,371,326	228,369 (1.2%)/19,142,957 (98.8%)	61,655	249,834
Pro_RBD_Male	31	12,264,624	12,264,624	63,331 (0.5%)/12,201,293 (99.5%)	61,898	250,173
Pro_RBD_Female	6	9,568,588	9,568,588	40,093 (0.4%)/9,528,495 (99.6%)	61,542	249,619
Pro_Hypo_Male	14	12,081,647	12,081,647	69,229 (0.6%)/12,012,418 (99.4%)	61,924	250,185
Pro_Hypo_Female	7	9,942,790	9,942,790	29,510 (0.3%)/9,913,280 (99.7%)	61,565	249,627

**Table 2 diagnostics-14-00929-t002:** Summary of the pathway and association disease analysis representing the major diseases detected from datasets of DisGeNET and HPO, along with the number of genes involved for them in each population.

Population	Disease	Number of Genes	Dataset
HC_Male	Acquired Hypogammaglobulinemia	11	DisGeNET
	Hyperinsulinemia	120	HPO
HC_Female	Dermatologic Disorders	75	DisGeNET
	Autosomal Dominant Inheritance	1828	HPO
Pro_Gen_Male	Pheochromocytoma	19	DisGeNET
	Cerebral Hemorrhage	62	HPO
Pro_Gen_Female	Single Seizure	101	DisGeNET
	Autosomal Dominant Inheritance	1828	HPO
Pro_RBD_Male	Cone–Rod Synaptic Disorder (CRSD)	13	DisGeNET
	Respiratory Insufficiency Due to Muscle Weakness	79	HPO
Pro_RBD_Female	Autosomal Recessive Primary Microcephaly	22	DisGeNET
	Autosomal Dominant Inheritance	1828	HPO
Pro_Hypo_Male	Arthritis; Adjuvant-Induced	40	DisGeNET
	Prenatal Maternal Abnormality	23	HPO
Pro_Hypo_Female	Familial Alzheimer’s Disease (FAD)	99	DisGeNET
	Autosomal Dominant Inheritance	1828	HPO

**Table 3 diagnostics-14-00929-t003:** Recently, 12 potentially novel PD genes were detected in the populations with prodromal PD (genetic male and female), prodromal PD (adolescent and female), and prodromal PD (hyposmia male and female).

	Population	Pro_Gen_Male Gene Count	Pro_Gen_Female Gene Count	Pro_RBD_Male Gene Count	Pro_RBD_Female Gene Count	Pro_Hypo_Male Gene Count	Pro_Hypo_Female Gene Count
Gene Name							
MTF2		2985	3458	3065	1397	2035	1606
ADD1		9095	10,457	5535	3471	4404	3722
PIK3CA		2958	3072	1874	1390	1606	1830
SYBU		13,897	15,607	7991	5943	8012	6030
IRS2		237	291	192	138	210	147
USP8		5997	6524	4432	3358	4953	1953
PIGL		8277	9021	4350	2819	3902	2923
FASN		1184	1224	691	552	775	593
MYLK2		265	296	183	132	111	107
USP25		4081	4952	2672	753	2796	534
EP300		3146	3510	1704	1256	1790	1398
PPP6R2		5652	6393	3090	2693	3723	2561

## Data Availability

Use of artificial intelligence tools: During the preparation of this work, AI tools were used to improve the readability and language of the manuscript and to generate images, and subsequently, the authors revised and edited the content produced by the AI tools as necessary, taking full responsibility for the ultimate content of the present manuscript.

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
