# Peer review of "Novel Variants Linked to the Prodromal Stage of Parkinson’s Disease (PD) Patients"

_diagnostics, 2024, doi:10.3390/diagnostics14090929_

Round 1

Reviewer 1 Report

Comments and Suggestions for Authors

The authors herein present an interesting research using VCG analysis for the detection of novel variants and genes linked to the prodromal stage of PD by making use of data from PPMI encompassing a wide range of participants belonging to different groups. Percentages of novel variants were determined in the prodromal subgroups and 12 novel loci recently found in PD patients were observed in the prodromal phase of disease.  These could potentially be used as prognostic indicators in the early diagnosis of prodromal PD.
The overall research methodology is extensive and authors have maximized the advantages of all novel tools for collecting data from the PPMI. The results are well presented and discussed in detail. The overall findings are undoubtedly useful for the scientific community and certainly point towards the advancement of knowledge towards the early detection of prodromal PD.

Some very minor suggestions are given below and these can be easily addressed by the authors following which the revised manuscript can be considered for acceptance.

1.        It would be beneficial to provide some more background on PPMI in the introductory section. Section 3.1 shows some statements, but it would be preferred to provide some more information in the beginning for context.

2.        Figures 2, 4-6: please consider improving the resolution of current figures – especially to see if font sizes can be increased for clarity and ease of understanding.

3.        Page 5, lines 141-145: it is encouraged to provide literature support/ references for these lines.

4.        Sections 5 and 6: consider providing the gist of these sections- for eg: conditions detected in ‘x’ population and numbers in a table so that it can improve readability. The current paragraphs are wordy and may distract readers. The content is definitely appreciated- but would be great to make it concise and include tabular formats for ease of reading.  

Author Response

We would like to thank the reviewer for the contructive feedback

Reviewer 2 Report

Comments and Suggestions for Authors

1.       In methods clarify whether the April 2021 cohort is representative of the general PD population or if it has specific characteristics that could influence the findings.

2.       The manuscript mentions the use of gVCF data but does not specify the versions of the bioinformatics tools or the parameters used for the analyses, such as the thresholds for filtering variants.

3.       The selection criteria and characteristics of the healthy control group need to be more clearly defined. It is essential to ensure that controls are appropriately matched not only in terms of health status but also genetic background. This is crucial for the accuracy of comparative analyses, as unrecognized population stratification could lead to misleading conclusions about genetic associations with prodromal PD.

4.       “All female participants in this study had a Y chromosome”. This is a major limitation of the study, as the findings representing “female” cannot be applied on genetic females (XX), which represent the majority of females in the world. This investigation studies genetic males only. This is a very interesting finding and throughout the manuscript the two groups can be referred to as AIS vs no AIS instead of male vs female since genetically speaking both groups are XY.

5.       Shouldn’t figures 4, 5, and 6 be included in the supplemental material rather the manuscript itself? How do these figures aid our understanding? The quality of the images is not good, meaning when I zoom in to read the annotations of the yellow and blue dots, it is unclear what the text says. The significance of the colors red and pink is not clear because of the low quality of the images. Explanation behind each of these colors could be included in the figure legend. Better quality of the images should be provided regardless of the author’s decision to keep the images in the manuscript or move them to the supplemental material file.

6.       In “Novel Gene Detection in Prodromal PD Populations”. An analysis and discussion regarding the difference between the count of these genes among the groups could be done. For example, which of the genes is most highly associated with prodormal RBD? Are there other studies which support the association of these genes with the symptoms seen among these individuals?

7.       In section “Disease Gene Network (DisGeNET) Detection in Prodromal PD Populations”, there’s a gap between detecting gene variants associated with diseases and using this information to influence treatment or prevention strategies.

8.       The manuscript would benefit from a discussion on how the identified genetic markers could be used clinically. Are there potential diagnostic or therapeutic implications of these markers & How might these findings lead to personalized medicine approaches in PD?

9.       The study involves different prodromal subgroups (genetic, RBD, hyposmia). It would be useful to see a subgroup analysis to understand if certain genetic variants are more associated with specific prodromal symptoms. This could help clarify the pathophysiological pathways leading to diverse manifestations of PD.

Author Response

We would like to thank the reviewer for the constructive feedback

Round 2

Reviewer 2 Report

Comments and Suggestions for Authors

Thank you for taking all comments into consideration.